

# Response of growth and physiological enzyme activities in *Eriogyna pyretorum* to various host plants

Haoyu Lin[1,2], Songkai Liao[1], Hongjian Wei[1], Qi Wang[1], Xinjie Mao[1], Jiajin Wang[1], Shouping Cai[2] and Hui Chen[1]

[1] State Key Laboratory of Conservation and Utilization of Subtropical Agro Bioresources, Guangdong Laboratory for Lingnan Modern Agriculture, College of Forestry and Landscape Architecture, South China Agricultural University, Guangzhou, Guangdong, China
[2] Fujian Academy of Forestry, Fuzhou, Fujian, China

## ABSTRACT

Morphological attributes and chemical composition of host plants shape growth and development of phytophagous insects via influences on their behavior and physiological processes. This research delves into the relationship between *Eriogyna pyretorum* and various host plants through studuying how feeding on different host tree species affect growth, development, and physiological enzyme activities. We examined *E. pyretorum* response to three distinct host plants: *Camphora officinarum*, *Liquidambar formosana* and *Pterocarya stenoptera*. Notably, larvae feeding on *C. officinarum* and *L. formosana* displayed accelerated development, increased pupal length, and higher survival rates compared to those on *P. stenoptera*. This underlines the pivotal role of host plant selection in shaping the *E. pyretorum*'s life cycle. The activities of a-amylase, lipase and protective enzymes were the highest in larvae fed on the most suitable host *L. formosana* which indicated that the increase of these enzyme activities was closely related to growth and development. Furthermore, our investigation revealed a relationship between enzymatic activities and host plants. Digestive enzymes, protective enzymes, and detoxifying enzymes exhibited substantial variations contingent upon the ingested host plant. Moreover, the total phenolics content in the host plant leaves manifested a noteworthy positive correlation with catalase and lipase activities. In contrast, a marked negative correlation emerged with glutathione S-transferase and α-amylase activities. The total developmental duration of larvae exhibited a significant positive correlation with the activities of GST and CarE. The survival rate of larvae showed a significant positive correlation with CYP450. These observations underscore the insect's remarkable adaptability in orchestrating metabolic processes in accordance with available nutritional resources. This study highlights the interplay between *E. pyretorum* and its host plants, offering novel insights into how different vegetation types influence growth, development, and physiological responses. These findings contribute to a deeper comprehension of insect-plant interactions, with potential applications in pest management and ecological conservation.

Corresponding authors
Shouping Cai, caishouping@163.com
Hui Chen, chenhuiyl@163.com

## INTRODUCTION

*Eriogyna pyretorum* (Westwood) (Lepidoptera: Saturniidae) is widely distributed in Myanmar, China, India, Malaysia, and Vietnam, and is notorious for its defoliating behavior (*Lin et al., 2023*). *E. pyretorum* exhibits distinct characteristics such as a short outbreak cycle, high feeding capacity, and prolific egg-laying, directly impacting the growth of ornamental and economically significant tree species, resulting in considerable economic losses and ecological damage (*Zhou et al., 2021*). In China, *E. pyretorum* poses a significant threat to 22 different plant species, including avenue trees and economically important trees such as *Cinnamomum officinarum*, *Liquidambar formosana*, *Pterocarya stenoptera*, and *Juglans regia* (*Li, Chen & Kang, 1990*). Specifically, in Fujian province and Jiangxi province of China, *E. pyretorum* primarily targets *C. officinarum*, with its larvae predominantly feeding on the leaves, resulting in reduced photosynthesis. This feeding behavior leads to economic losses and adversely affects the normal growth and development of the plants (*Yin et al., 2008*). In Zhejiang province, *E. pyretorum* mainly inflicts harm on *L. formosana* and *P. stenoptera* (*Zhou et al., 2021*). With the intensification of global climate change and the rapid development of transportation and logistics in recent years, the developmental processes of certain insects have been altered, resulting in shortened pest occurrence cycles and accelerated spread, exacerbating the harm caused by these insects to their host plants (*Moir et al., 2014*). Therefore, the development of effective control strategies to protect and mitigate the damage caused by *E. pyretorum* to its host plants is of paramount importance for China's greening efforts and ecological balance.

Plant secondary metabolites, including alkaloids, phenols, terpenes, and others, are well-known for their pivotal role in providing resistance against insects and other pests, acting as a reservoir for adapting to environmental changes and promoting plant development (*Razzaq et al., 2023*). These compounds possess toxic properties that can lead to the demise of insects upon ingestion and also serve as an indirect form of protection by attracting natural predators of the insects (*Arnold, Kruuk & Nicotra, 2019*). *Gitonga et al. (2022)* found a positive correlation between tannin content in cowpea (*Vigna unguiculata*) and resistance to flower bud thrips (*Megalurothrips sjostedti*). Similarly, in tomato (*Solanum lycopersicum*) plants, the presence of flavonoids exhibited inhibitory effects on oviposition by *Bemisia tabaci*, and varieties with high flavonoid content showed repellent effects against *B. tabaci* (*Kang et al., 2010*). Successful utilization of host plants by phytophagous insects depends on efficient nutrient digestion and defense from toxic and antinutritive secondary metabolites. They rely on their internal digestive enzymes to convert the nutritional components in plants into the energy required for their survival, growth and reproduction (*Sun et al., 2022*). The activity of digestive enzymes varies among insects with different feeding habits, and even insects with the same feeding habits may exhibit variations in enzyme activities depending on the specific host species they consume (*Zhang et al., 2023*). Phytophagous insects possess the ability to metabolically adapt to the secondary metabolites present in their host plants. They achieve this by utilizing detoxifying enzymes, such as carboxylesterases (CarE), glutathione S-transferases (GST), and multifunctional oxidases, to transform toxic secondary metabolites into non-toxic substances. This adaptation

enables insects to thrive on their host plants, making them suitable sources of sustenance (*Senthil-Nathan, 2013*; *Chen et al., 2015*; *Liu et al., 2016*; *Sun et al., 2019*). Additionally, protective enzymes within the insect's body, such as superoxide dismutase (SOD), catalase (CAT), and peroxidase (POD), play a crucial role in its resistance to various environmental stressors (*Wang et al., 2020*).

The variation in nutrient composition and secondary metabolite content among different host plants can lead to changes in enzyme activities (*e.g.*, digestive enzymes, protective enzymes, and detoxifying enzymes) within phytophagous insects. Previous studies have demonstrated that protease activity in the larvae of *Spodoptera litura* feeding on *Glycine max* is higher than those feeding on *Brassica rapa* and *Zea mays* (*Zhang, Li & Wu, 2013*). Similarly, the activities of acetylcholinesterase (AChE), GST, CarE, and insect cytochrome P450 (CYP450) in the larvae of *Spodoptera frugiperda* feeding on different host plants were significantly different (*Lu et al., 2020*). Moreover, the carbohydrate content in the host plant can influence the digestive enzyme activities in *Helicoverpa armigera* (*Kotkar et al., 2009*). *Borzoui et al. (2018)* demonstrated that the dietary carbohydrate and protein content induce changes in nutritional efficiency, development, and alpha-amylase activity of *Plodia interpunctella*. Therefore, a comprehensive exploration of the impact of different host plants on *E. pyretorum*'s enzyme activities and understanding its adaptation mechanisms to host plants are of significant importance for controlling this pest, ensuring the healthy growth of greenery and economically important tree species.

In this study, three different tree species (*C. officinarum*, *L. formosana* and *P. stenoptera*) were used as host plants. Newly hatched *E. pyretorum* larvae were fed with leaves from these plants, and their biological parameters (larval period, percentage pupation, larval width of the capsule, larval mass and length) were recorded. Subsequently, the insects were reared until the fourth instar, and the variations in digestive enzymes (lipase, α-amylase, and trypsin), protective enzymes (SOD, CAT, and POD), and detoxifying enzymes (CYP450, GST, and CarE) activities were assessed. Additionally, the relationship between enzyme activities and the content of secondary metabolites and nutrients in the host plant leaves was analyzed. The primary objective was to elucidate how feeding on different host plants influences the enzyme activities in *E. pyretorum*. The findings of this research will serve as a foundational basis for further investigating the host selection and adaptation mechanisms of *E. pyretorum* and developing effective pest management strategies.

## MATERIALS AND METHOD

### Insect collection and rearing

*E. pyretorum* pupae were collected from avenue trees in Fuzhou, Fujian province, and reared in the laboratory at $25 \pm 2$ °C and $60 \pm 10\%$ relative humidity (RH). On emergence, five pairs of adults were transferred to mating cages (50 cm height, 30 cm length and width) containing a potted camphor tree to mate and lay eggs. Using a small spoon, the eggs were carefully scraped off from the branches and leaves and gather them into a plastic rearing box (17 cm height, 10 cm length, and seven cm width). Larvae (<24 h old) were reared on different hosts after hatching in order to study the biology parameters.
### Host plant

*E. pyretorum* larvae are fed on *C. officinarum*, *L. formosana* and *P. stenoptera*, and the leaves of the three host species are fed to the larvae until pupation. The larvae are reared for one generation in each host species. Each replicate consisted of 30 larvae, with three replicates for each host species. Fresh leaves obtained daily from each host plant on the campus of South China Agricultural University in Guangzhou, Guangdong, were placed in three rearing boxes, with 10 larvae introduced into each box. On the 2nd instars (sixth day), larvae were separated and stored separately on the corresponding host plant leaves in rearing boxes. Fresh leaves were provided daily until *E. pyretorum* pupation. The newly emerged adults were transferred to mating cages, and each host plant was repeated three times. In each cage, we housed three to five pairs of adults. We monitored the daily egg-laying activity of the adults and recorded the number of eggs laid. The experiment was conducted under constant laboratory conditions of $25 \pm 2\,°C$, $60 \pm 10\%$ RH, and a 12:12 h light/dark cycle, and repeated three times.

### Evaluation of biological parameters on three different hosts

Observations were taken each day to evaluate the growth and development of *E. pyretorum* on different hosts. Biological parameters—larval period, percentage molt, and percentage pupation—were observed for *E. pyretorum* fed on its natural hosts for one successive generation. Molted head capsules were identified to determine the progress of metamorphosis in the larval stages and the length of the larva was measured after each molt. Exuvial larval head capsules of all instars were collected after each molt, and the width of the capsule measured using a microscope reticule ($\times 8$, Leica MZ6 modular stereomicroscope). Larval weight and length were measured on day 2 after each molt. Pupal weight was measured 14 d after pupation (cut the cocoon before measuring). Percent mortality was calculated for each host and diet by dividing the number of dead individuals by the total number of individuals used in the study. Prepupal stage means the stage in which the last instar larvae of the *E. pyretorum* stop eating and curl up their bodies before pupation.

Larvae survival rate (%) = number of next instar larvae/number of previous instar larvae

Pupae survival rate (%) = number of pupae/number of prepupal stage

Total larvae survival rate (%) = number of pupae/number of rearing larvae

### Determination of digestive enzymes, protective enzymes and detoxification enzymes

The 4th instar larvae (on the second day) of each *E. pyretorum* host were transferred into a five mL centrifuge tube. Phosphate buffer solution with a pH of 7.0 and a concentration of 0.1 mol/L (containing 1% polyvinylpyrrolidone) was added to the tube, followed by homogenization under ice bath conditions. The homogenate was then centrifuged at 8,000 g, 0–4 °C for 10 min, and the resulting supernatant was used as the enzyme solution for testing. Enzyme activity assays for lipase, $\alpha$-amylase, trypsin, SOD, CAT, POD, CYP450, GST, and CarE were conducted following the instructions provided by Nanjing Jiancheng Bioengineering Research Institute. Each experimental group was replicated three times, with one larva per replicate.

## Determination of nutrient content and secondary metabolites in the leaves of host plants

Different host plant leaves were washed with deionized water, cut into small pieces with scissors, and divided for further measuring. Soluble sugars and soluble proteins, as well as three secondary metabolites—tannins, flavonoids, and total phenols—were determined using the Plant Soluble Sugar Content Assay Kit (KT-2-Y; Suzhou Comin Biotechnology Co., Ltd., Suzhou, China), BCA Protein Content Assay Kit (BCAP-2-W; Suzhou Comin Biotechnology Co., Ltd.), Tannin Content Assay Kit (DN-2-Y; Suzhou Comin Biotechnology Co., Ltd.), Plant Flavonoid Test Kit (LHT-2-G; Suzhou Comin Biotechnology Co., Ltd.), and Plant Total Phenol Test Kit (TP-1-G; Suzhou Comin Biotechnology Co., Ltd.), respectively. Five samples (three leaves for a sample) were taken for each substance content determination, with each sample measured in triplicate.

## Statistical analysis

Larval period, larval mass, larval length, head capsule width, pupal length, pupal width, and survival rates were calculated for individuals fed on three host plants. Fisher's LSD test was conducted to determine significant differences in the ANOVA for biological parameters and physiological enzymes (protective, detoxification and digestive enzymes) activities. We determined Pearson's product moment correlation coefficients ($P < 0.05$) between the activities of protective, detoxification and digestive enzymes as well as between the growth status and enzyme activities in *E. pyretorum*. We drew correlation heatmaps using Origin 2021. The statistical analyses were conducted using SPSS version 26.0, and bar graphs were generated using GraphPad Prism version 7.

## RESULTS

### Growth status of *E. pyretorum* reared on different host plants

*E. pyretorum* larvae were able to complete their growth and development on three different host plants, but there were differences in the developmental periods. Larvae fed on *L. formosana* and *P. stenoptera* underwent 7th instar, while those fed on *C. officinarum* experienced 8th instar. Specifically, the 2nd, 4th and 6th instars of *C. officinarum*-fed larvae had significantly shorter developmental periods compared to larvae feeding on *P. stenoptera* ($P < 0.05$). The 4th and 6th instars of larvae feeding on *P. stenoptera* exhibited significantly longer development duration compared to those feeding on the other two host plants. And the *E. pyretorum* larvae feeding on *C. officinarum* exhibited significantly longer total larval development duration compared to those feeding on the other two host plants ($P < 0.05$). However, there were no significant differences in the development duration of the 1st, 5th, and 7th instars among larvae feeding on the three host plants ($P > 0.05$) (Fig. 1B).

Moreover, there were differences in the mass and length of *E. pyretorum* larvae fed on the three host plants. The 5th and 7th instars showed significant differences in mass and length among the larvae: *L. formosana* > *C. officinarum* > *P. stenoptera* ($P < 0.05$) (Figs. 1A, 1C). The length of 1st, 2nd, 3rd, and 4th instars larvae feeding on *P. stenoptera* was also significantly shorter than those feeding on the other two host plants ($P < 0.05$), while there were no significant differences in length among larvae feeding on *L. formosana* and

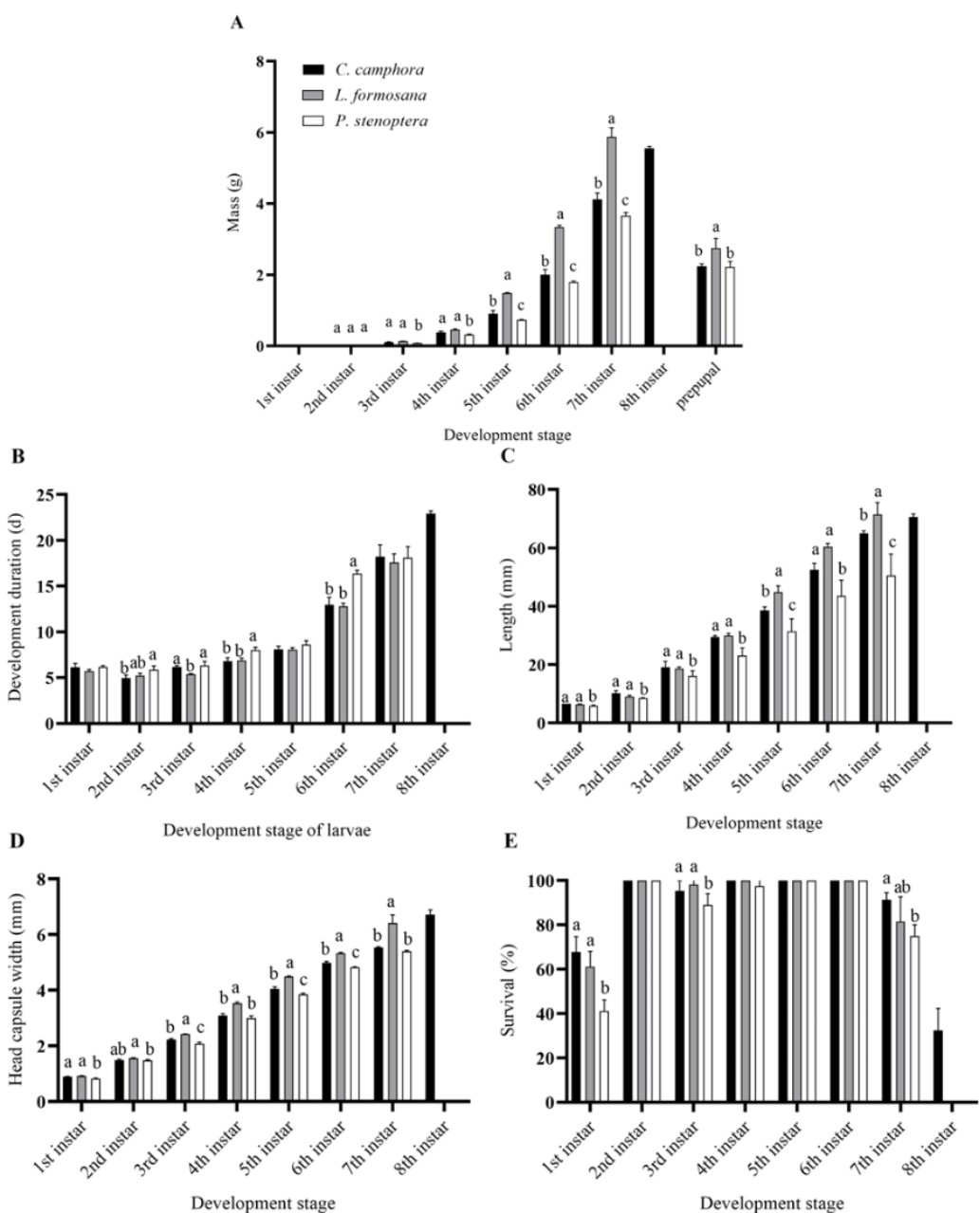

**Figure 1** Biology parameters including (A) mass, (B) development duration, (C) length, (D) head capsule width, (E) survival, of *Eriogyna pyretorum* larvae reared on different host plants. *C. officinarum*, reared on *C. officinarum*; *L. formosana*, reared on *L. formosana*; *P. stenoptera*, reared on *P. stenoptera*. All data are expressed as means ± standard deviations ($n = 3$). The different letters above the columns indicate a significant difference among the means according to Fisher's LSD test ($P < 0.05$).

*C. officinarum* ($P > 0.05$). Additionally, the mass of 3rd and 4th instars larvae feeding on *P. stenoptera* was significantly lower than those feeding on the other two host plants ($P < 0.05$), while there were no significant differences in mass among larvae feeding on *L. formosana* and *C. officinarum* ($P > 0.05$). Furthermore, the study found that *E. pyretorum*

larvae feeding on *L. formosana* had the highest mass in each instar after the 2nd instar, while those feeding on *P. stenoptera* had the lowest mass, indicating that *L. formosana* is more suitable for the nutrient accumulation and growth of *E. pyretorum*.

In terms of head capsule width, the 3rd, 4th, 5th, 6th, and 7th instars larvae feeding on *L. formosana* had significantly larger head capsule widths than those feeding on the other two host plants, while the 3rd, 5th and 6th instars larvae feeding on *P. stenoptera* had significantly smaller head capsule widths than those feeding on *C. officinarum* and *L. formosana* (Fig. 1D). Furthermore, the survival of 7th instar larvae feeding on *C. officinarum* had significantly higher than those feeding on *P. stenoptera* (Fig. 1E).

The pupal length of *E. pyretorum* feeding on *L. formosana* and *C. officinarum* showed no significant difference ($P > 0.05$) but was significantly greater than those feeding on *P. stenoptera* ($P < 0.05$). However, the pupal width of *E. pyretorum* feeding on *L. formosana* was significantly higher than those feeding on the other two host plants ($P < 0.05$) (Fig. 2A). Moreover, the survival rates of *E. pyretorum* feeding on *L. formosana* and *C. officinarum* exhibited no significant difference ($P > 0.05$) but were significantly higher than those feeding on *P. stenoptera* ($P < 0.05$) (Fig. 2B). The total larvae duration of larvae feeding on *C. officinarum* exhibited significantly longer development duration compared to those feeding on other two host plants (Fig. 2C).

## Effects of feeding on different host plants on the activity of digestive enzymes in *E. pyretorum* larvae

The larvae of *E. pyretorum*, when feeding on leaves of different host plants, exhibited variations in digestive enzyme activities. Specifically, the $\alpha$-amylase activity in *E. pyretorum* larvae feeding on *P. stenoptera* leaves was significantly lower than that in larvae feeding on *L. formosana* and *C. officinarum* leaves ($P < 0.05$), while no significant difference was observed in α-amylase activity between larvae feeding on *L. formosana* and *C. officinarum* leaves ($P > 0.05$) (Fig. 3A). Regarding Trypsin activity, *E. pyretorum* larvae fed on the three host plants followed the order: *P. stenoptera* > *C. officinarum* > *L. formosana*, with all differences being statistically significant ($P < 0.05$) (Fig. 3B). Furthermore, the lipase activity in *E. pyretorum* larvae feeding on *L. formosana* leaves was significantly higher than that in larvae feeding on *C. officinarum* and *P. stenoptera* leaves, while no significant difference in lipase activity was observed between larvae feeding on *C. officinarum* and *P. stenoptera* leaves ($P > 0.05$) (Fig. 3C).

## Effects of feeding on different host plants on the activity of detoxification enzymes and protective enzymes in *E. pyretorum*

Differences were observed in the detoxifying enzyme activities of *E. pyretorum* larvae feeding on leaves of different host plants. Specifically, the CarE and CYP450 activities in *E. pyretorum* larvae feeding on *C. officinarum* leaves were significantly higher than those in larvae feeding on *L. formosana* and *P. stenoptera* ($P < 0.05$) (Figs. 4A, 4B). However, no significant differences were found in CarE and CYP450 activities between larvae feeding on *L. formosana* and *P. stenoptera* leaves ($P > 0.05$). Additionally, the GST activity in *E. pyretorum* larvae feeding on *L. formosana* leaves was significantly lower than that in larvae feeding on *C. officinarum* and *P. stenoptera* leaves (Fig. 4C). As for protective

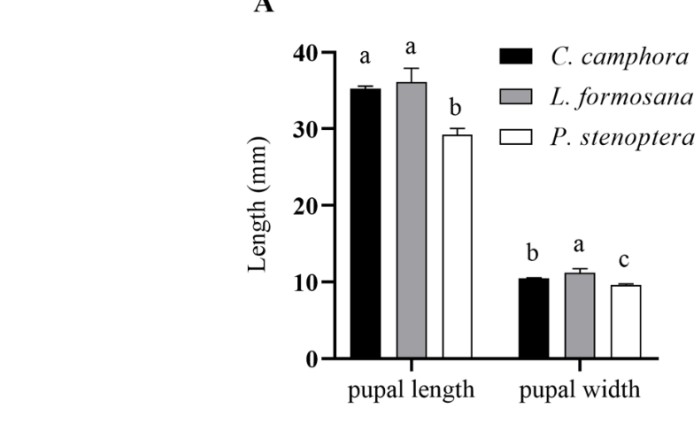

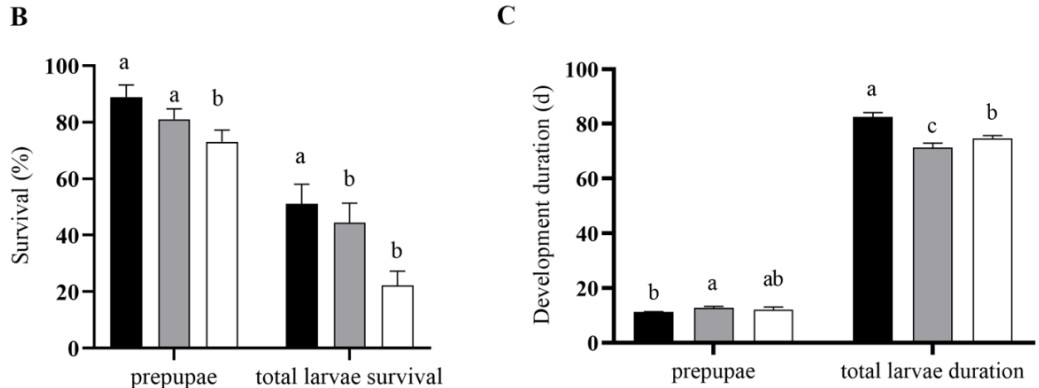

**Figure 2 Pupal length and pupal width (A), pupae and larvae survival rate (B), development duration (C) of *Eriogyna pyretorum* larvae reared on different host plants.** *C. officinarum*, reared on *C. officinarum*; *L. formosana*, reared on *L. formosana*; *P. stenoptera*, reared on *P. stenoptera*. All data are expressed as means ± standard deviations (*n* = 3). The different lowercase letters above the columns indicate a significant difference among the means according to Fisher's LSD test (*P* < 0.05).

enzymes, there were no significant differences in SOD activity among *E. pyretorum* larvae feeding on the three host plants (*P* > 0.05) (Fig. 4D). However, the POD and CAT activities in *E. pyretorum* larvae feeding on the three host plants followed the order: *L. formosana* > *C. officinarum* > *P. stenoptera*, and the differences were statistically significant (*P* < 0.05) (Figs. 4E, 4F).

## Nutrient content and secondary metabolites in the leaves of three host plants

There were no significant differences in soluble sugar content among the leaves of the three host plants (*P* > 0.05). However, the soluble protein content in *C. officinarum* leaves was the highest, measuring 2.58 ± 0.34 mg/g, which was significantly higher than the other two host plants (*P* < 0.05). Specifically, the soluble protein content in *C. officinarum* leaves was 1.28 times that of *L. formosana* leaves and 2.48 times that of *P. stenoptera* leaves (Table 1).

There were no significant differences in flavonoid content among the leaves of the three host plants (*P* > 0.05). The total phenolics content in *L. formosana* leaves was the highest,

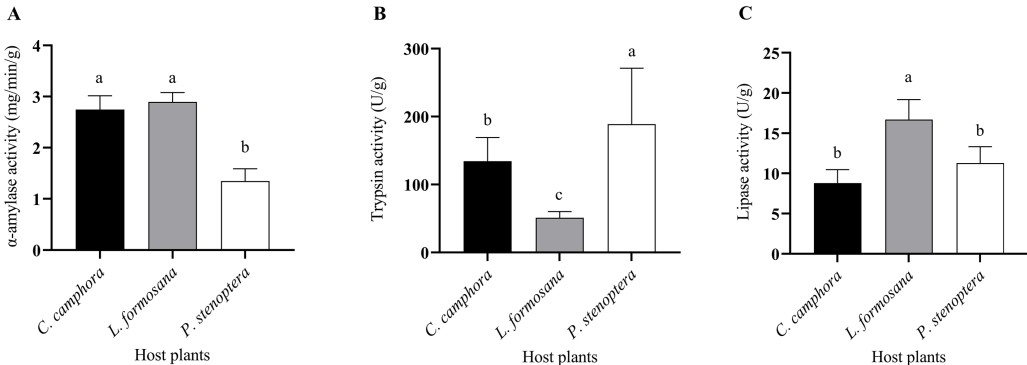

**Figure 3** **Digestive enzymes including (A) α-amylase, (B) trypsin, (C) lipase activities of *Eriogyna pyretorum* larvae reared on different host plants.** *C. officinarum*, reared on *C. officinarum*; *L. formosana*, reared on *L. formosana*; *P. stenoptera*, reared on *P. stenoptera*. All data are expressed as means ± standard deviations ($n = 3$). The different lowercase letters above the columns indicate a significant difference among the means according to Fisher's LSD test ($P < 0.05$).

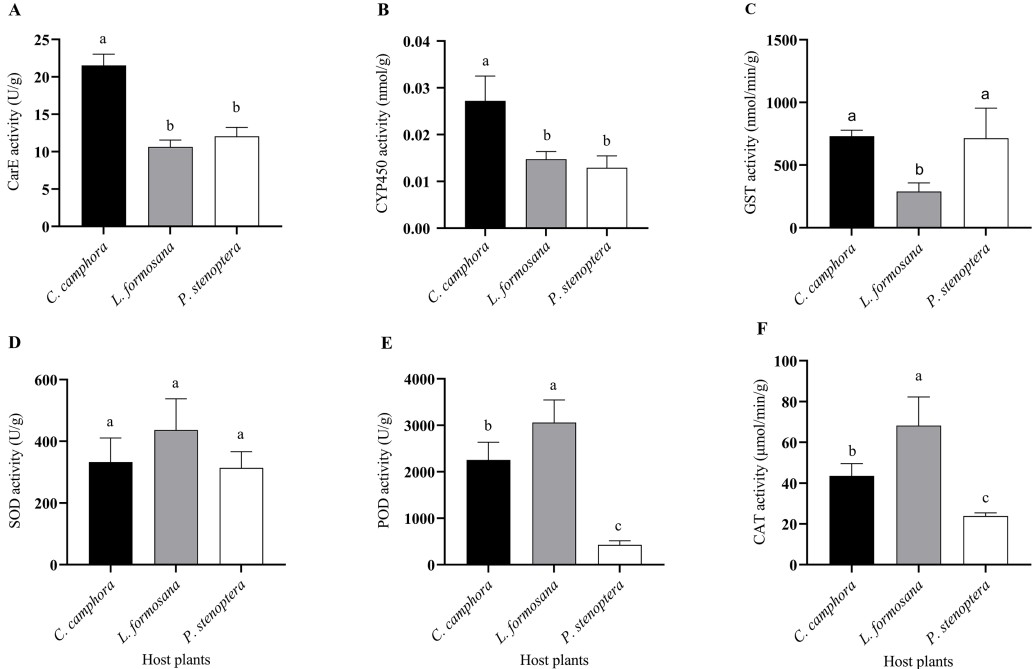

**Figure 4** **Detoxifying enzymes and protective enzymes including (A) CarE, (B) CYP450, (C) GST, (D) SOD, (E) POD, and (F) CAT activities of *Eriogyna pyretorum* larvae reared on different host plants.** *C. officinarum*, reared on *C. officinarum*; *L. formosana*, reared on *L. formosana*; *P. stenoptera*, reared on *P. stenoptera*. All data are expressed as means ± standard deviations ($n = 3$). The different lowercase letters above the columns indicate a significant difference among the means according to Fisher's LSD test ($P < 0.05$).

**Table 1 The contents of nutrients and contents of secondary metabolites in leaves of three host plants.** Different lowercase letters in the same column indicated significant difference ($P < 0.05$).

| | Nutrients | | Secondary metabolites | | |
|---|---|---|---|---|---|
| | Soluble sugar (mg/g) | Soluble protein (mg/g) | Total phenolics content (mg/g) | Tannin content (mg/g) | Flavonoid content (mg/g) |
| *C. officinarum* | 6.85 ± 0.37a | 1.90 ± 0.41a | 27.99 ± 1.23b | 1.17 ± 0.12a | 56.58 ± 6.00a |
| *L. formosana* | 6.72 ± 0.26a | 1.48 ± 0.54ab | 33.04 ± 0.89a | 1.20 ± 0.09a | 55.45 ± 4.56a |
| *P. stenoptera* | 6.43 ± 0.54a | 0.77 ± 0.21b | 28.35 ± 2.28b | 1.38 ± 0.02a | 53.49 ± 5.57a |

measuring 33.04 ± 2.57 mg/g, which was significantly higher than the other two host plants ($P < 0.05$). However, there were no significant differences in total phenolics content between *C. officinarum* and *P. stenoptera* leaves ($P > 0.05$). As for tannin content, *P. stenoptera* leaves had the highest content, reaching 1.38 ± 0.19 mg/g, which was significantly higher than the other two host plants ($P < 0.05$). Nonetheless, there were no significant differences in tannin content between *C. officinarum* and *L. formosana* leaves ($P > 0.05$).

## Correlation analysis between enzyme activities in *E. pyretorum* and contents of substances in leaves

The soluble protein content in host plant leaves exhibited a significant positive correlation with the activities of CYP450, POD, and trypsin in *E. pyretorum* larvae. However, no significant correlation was observed between the soluble sugar content in host plant leaves and the enzymatic activities in *E. pyretorum*. Notably, the total phenolics content in host plant leaves displayed a significant positive correlation with CAT and lipase activities in *E. pyretorum*, while demonstrating a significant negative correlation with GST and α-amylase activities. Furthermore, the tannin content in host plant leaves showed a significant negative correlation with CAT activity in *E. pyretorum*, whereas the flavonoid content in host plant leaves exhibited a significant positive correlation with trypsin activity in *E. pyretorum* (Fig. 5).

## Correlation analysis between enzyme activities in *E. pyretorum* and growth status of *E. pyretorum*

The total developmental duration of *E. pyretorum* larvae exhibited a significant positive correlation with the activities of GST and CarE, indicating that detoxifying enzymes have a notable impact on the total developmental duration of larvae. The survival rate of larvae showed a significant positive correlation with CYP450, suggesting that CYP450 influences larval mortality. The length and weight of 7th instar larvae were significantly positively correlated with POD activity. In contrast, CAT and trypsin activities were significantly negatively correlated with the length and weight of 7th-instar larvae as well as pupal length. These enzymes contribute to variations in the size of *E. pyretorum* larvae and pupae. However, no correlation was found between the activities of CYP450, CarE, α-amylase, and the body length/weight of larvae and pupae (Fig. 6).

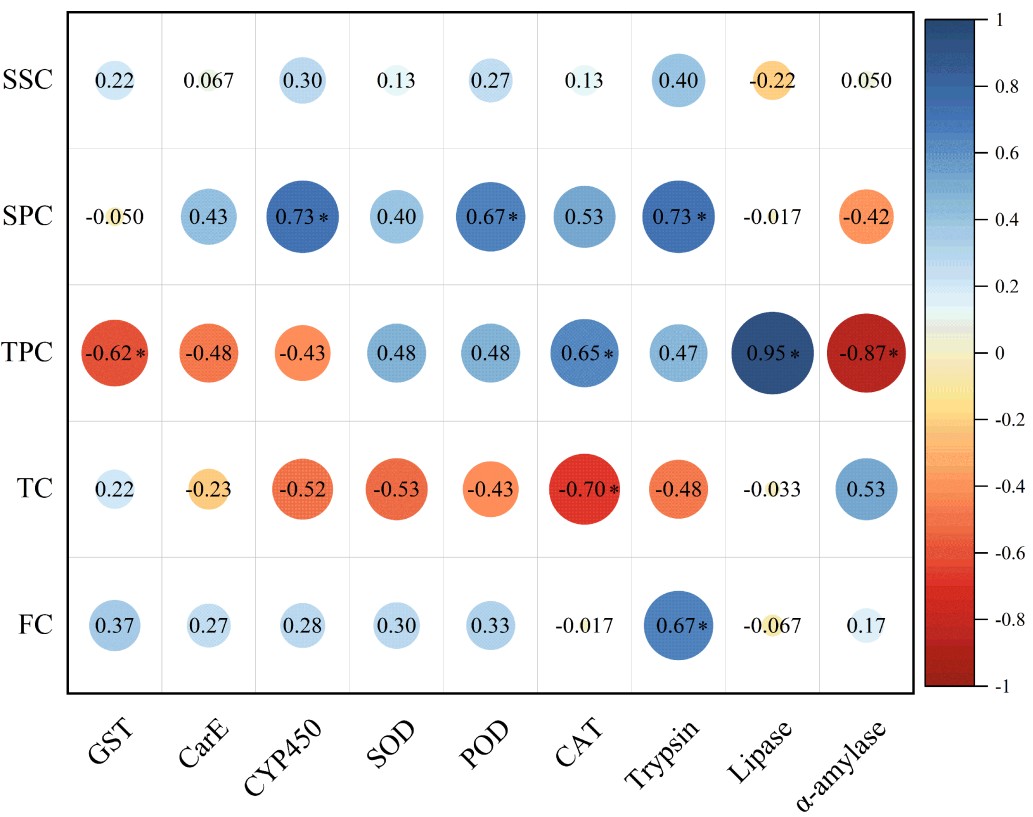

**Figure 5** Pearson's correlation coefficients between the activities of detoxifying enzymes, protective enzymes and digestive enzymes in *Eriogyna pyretorum* and the contents of secondary metabolites and nutrient substance. SSC, Soluble sugar content; SPC, soluble protein content; TPC, Total phenolics content; TC, tannin content; FC, flavonoid content. An asterisk (*) indicates significant correlation ($P < 0.05$).

## DISCUSSION

Feeding on different host plants profoundly influences the growth, development, and reproductive capacity of insects (*Fang, Qiao & Zhang, 2011*). For example, *Myzus persicae* exhibits considerable variations in larval development rate, developmental period, pupal duration, and pupal mass when feeding on different host plants (*Jiang et al., 2022*). A heavier pupal mass is generally indicative of greater adaptability, and a shorter developmental period serves as a measure of the suitability of host plants to phytophagous insects (*Leuck & Perkins, 1972*). In the context of this study, *E. pyretorum* larvae displayed a complete life cycle while feeding on *C. officinarum*, *L. formosana*, and *P. stenoptera*, confirming the species' polyphagous nature and remarkable ability to adapt to diverse host plants. Field surveys in regions such as East China, including Fujian and Zhejiang provinces, further validated the extensive damage inflicted on *Cinnamomum japonica* by *E. pyretorum* (*Yin et al., 2008*). Moreover, in Southwest China regions like Yunnan and Sichuan provinces, this pest posed a significant threat to various plants, including walnut (*Li, Chen & Kang, 1990*). These findings align with the research outcomes, collectively highlighting the

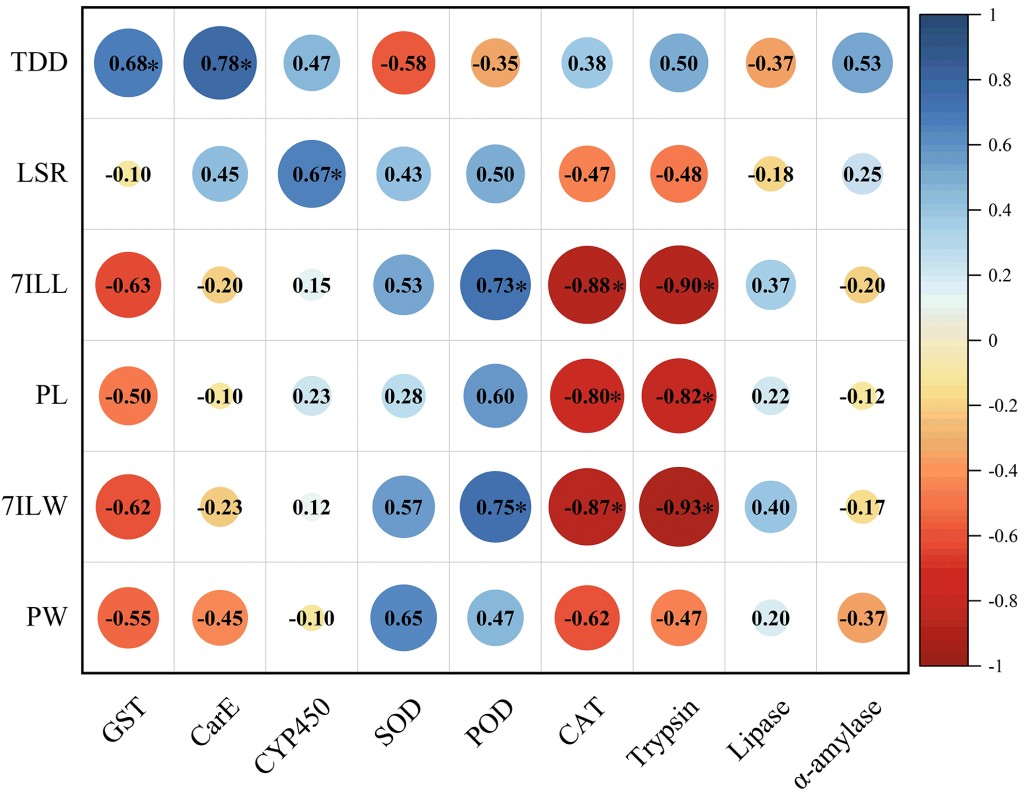

**Figure 6** Pearson's correlation coefficients between the activities of detoxifying enzymes, protective enzymes and digestive enzymes in *Eriogyna pyretorum* and the growth status. TDD, total developmental duration; LSR, larvae survival rate; 7ILL, 7th instar larval length; PL, pupal length; 7ILW, 7th instar larval weight; PW, pupal weight. An asterisk (*) indicates significant correlation ($P < 0.05$).

adaptive prowess of *E. pyretorum* in relation to a broad range of host plants. This study demonstrated that *E. pyretorum* larvae feeding on *C. officinarum* and *L. formosana* exhibit significantly shorter development duration, higher pupal length, and increased survival rates, suggesting that these two host plants may be more conducive to the growth and development of *E. pyretorum*. The adaptability of phytophagous insects to different host plants is likely linked to the morphological characteristics, nutritional composition, and secondary metabolites of the host plants (*Miao, Han & Zhang, 2014*), warranting further comprehensive investigation in this domain for a deeper understanding.

The host selection of phytophagous insects is closely associated with the evolution of their enzyme systems (*Karasov, Rio & Caviedes-Vidal, 2011*). In response to different host plants, phytophagous insects exhibit genetic adaptability, inducing the production of a diverse array of digestive and detoxification enzymes with broader applicability, stronger effects, and higher sensitivity (*Ragland et al., 2015*). Subsequently, these insects regulate the expression levels of digestive and detoxification enzymes to enhance nutrient utilization in their gut. Key enzymes involved in carbohydrate metabolism, such as amylase, glucosidase, and α-amylase, display differential expression in the insect's gut in response to feeding on different host plants (*Srinivasan, Giri & Gupta, 2006*). This suggests that

phytophagous insects can modulate their internal digestive enzyme activities based on their developmental needs when consuming different host plants. In this study, the lipase activity in *E. pyretorum* larvae feeding on leaves of *L. formosana* was significantly higher than in larvae feeding on leaves of other host plants. Conversely, the trypsin activity in larvae feeding on *P. stenoptera* leaves showed a significant increase. These differences may be attributed to the self-regulation of *E. pyretorum* in response to feeding on different host plants. When feeding on hosts with lower soluble protein content, *E. pyretorum* may enhance trypsin activity to obtain more proteins, which could contribute to the wide host range of *E. pyretorum* (*Montezano et al., 2018*).

Insects have evolved intricate metabolic adaptations to tolerate potential toxins in plants, enabling broader dietary options to enhance biodiversity and inhabit various environments (*Rotkopf et al., 2013*). SOD catalyzes the dismutation of superoxide anion radicals, while POD eliminates hydrogen peroxide and toxic substances, such as phenols and amines, in animals and plants, participating in the metabolism of reactive oxygen species. CAT converts excessive hydrogen peroxide ($H_2O_2$) into water and oxygen, providing a protective role in organisms (*Inayat et al., 2022*; *Wei et al., 2023*; *Yuan et al., 2021*). Plant secondary metabolites can induce the generation of a large quantity of superoxide anion radicals in the insect's body, thereby activating the activity of protective enzymes to safeguard the insect from oxydative damages to macromolecules (*Després, David & Gallet, 2007*; *Piskorski & Dorn, 2011*). The results of this study indicate that *L. formosana* leaves contain markedly higher levels of total phenolics compared to the other two host plants tested, consistent with the findings of *Taggar & Gill (2016)* and *Després, David & Gallet (2007)*. Additionally, there were no significant differences observed in flavonoid levels among larvae feeding on the three host plants. As a consequence of feeding on *L. formosana* leaves, *E. pyretorum* larvae show significantly elevated activities of POD and CAT within their bodies. This observation suggests that the secondary metabolites present in *L. formosana* leaves activate the protective enzymes in *E. pyretorum*, leading to a more effective detoxification of these secondary metabolites. The high activity of lipase in larvae fed on *L. formosana* leaves could indicate an enhanced ability of larvae to digest lipids, potentially providing them with a nutritional advantage that outweighs any negative effects of the high phenolic content. The positive correlation between total phenolics of host leaves and CAT/lipase activities also proves this point.

Host plants also exert an impact on the activity of detoxifying enzymes. GST, CarE and CYP450 are essential detoxifying enzymes in insect, playing pivotal roles in the breakdown of exogenous toxins and maintaining normal physiological metabolism (*Bai et al., 2023*; *Kumar et al., 2023*). The induction of CYP450 and GST by allelochemicals, such as terpenoids, flavonoids, and alkaloids, has also been reported (*Liu et al., 2023*). In the research conducted by *Xu et al. (2018)*, adult leaf beetles, *Monolepta hieroglyphica*, exhibited minor fluctuations in the activity of three detoxifying enzymes after feeding on four relatively preferred host plants, *Gossypium hirsutum*, *Zea mays*, *Glycine max* and *Setaria italica*. However, a significant increase in GST and CarE activities was observed in their bodies after feeding on two less suitable host plants, namely tomato and *Trapa natans*. In contrast to the aforementioned research, *E. pyretorum* larvae feeding on *C. officinarum*

exhibited the highest detoxifying enzyme activity. However, despite this elevated enzymatic activity, *C. officinarum* did not result in the poorest larval performance among the three host plants studied. Phenolic compounds are widely recognized for their importance in plant health, with numerous studies indicating that insect feeding can increase their levels (*Wallis & Galarneau, 2020*). The activity of phenolic compounds is closely linked to plant defense mechanisms. The alkaline pH in the gut of lepidopteran larvae facilitates the oxidation of polyphenols and inhibits their protein-precipitating function (*Salminen & Karonen, 2011*). In response, insect herbivores can utilize ingested ascorbate to reduce the oxidative activity of polyphenols, detoxify flavonoid aglycones through glycosylation, or passively excrete the metabolites they consume (*Barbehenn, Weir & Salminen, 2008*; *Salminen et al., 2004*). The results of this study indicate that *L. formosana* exhibits significantly higher levels of phenolic compounds compared to the other two host plants. However, *L. formosana* is not the least favorable for larval growth and development among the three host plants tested. This suggests that different insects have distinct mechanisms for metabolizing secondary metabolites in host plants, highlighting the complexity of these processes.

Interestingly, the total phenolics, tannins, and flavonoid content in the host plant leaves used for feeding *E. pyretorum* larvae showed little significant correlation with the three enzyme activities (except for a correlation with total phenolics and GST). This implies that tannins and flavonoids may not be the primary factors inducing an increase in detoxifying enzyme activities in *E. pyretorum* larvae. The notable inverse relationship between total phenolic content and GST activity might suggest an inhibitory effect of phenolics on GST activity. This inhibitory phenomenon has also been observed in insects such as *Micromelalopha troglodyta* and *Clostera anachoreta* (*Tang et al., 2011*; *Tang et al., 2014*; *Gawande & Khambalkar, 2014*).

However, the molecular mechanisms underlying *E. pyretorum*'s adaptation to different host plants and its adaptation mechanisms through intestinal microbial diversity warrant further investigation, and its adaptation mechanisms warrant further investigation which should also include intestinal microbial diversity.

## CONCLUSIONS

*E. pyretorum* exhibits remarkable physiological plasticity on different host plants, allowing it to regulate its internal digestive enzyme activities to efficiently utilize the varying nutrient contents of different host plants. Moreover, it enhances its adaptability to secondary metabolites or toxic substances in host plants by adjusting the activities of protective and detoxifying enzymes. This adaptive response enables *E. pyretorum* to expand its host spectrum. The significant physiological plasticity displayed by *E. pyretorum* is likely one of the key factors contributing to its rapid spread and establishment as an important invasive pest in China. Further study is needed to elucidate the molecular biology mechanisms underlying the responses of *E. pyretorum* to different host plants, as well as the mechanisms by which gut microbiota diversity enables adaptation to various hosts. Additionally, the impact of secondary metabolites on the gene expression in *E. pyretorum*, potentially altering it signaling transduction and energy metabolism pathways, requires further research to ascertain whether these metabolites exert effects similar to other toxic compounds.

## ACKNOWLEDGEMENTS

We express our deep gratitude to Professor Guanghong Liang, from Fujian Agriculture and Forestry University, Fuzhou, China, who provided *E. pyretorum* for this study.

### Funding
This research was funded by the National Natural Science Foundation of China (project number: 32071639) and the Laboratory for Lingnan Modern Agriculture Project (project number: NZ2021025). The funders had no role in study design, data collection and analysis, decision to publish, or preparation of the manuscript.

### Grant Disclosures
The following grant information was disclosed by the authors:
National Natural Science Foundation of China: project number: 32071639.
Laboratory for Lingnan Modern Agriculture Project: project number: NZ2021025.

### Competing Interests
The authors declare there are no competing interests.

### Author Contributions

- Haoyu Lin conceived and designed the experiments, analyzed the data, prepared figures and/or tables, and approved the final draft.
- Songkai Liao conceived and designed the experiments, authored or reviewed drafts of the article, and approved the final draft.
- Hongjian Wei conceived and designed the experiments, analyzed the data, prepared figures and/or tables, and approved the final draft.
- Qi Wang performed the experiments, prepared figures and/or tables, and approved the final draft.
- Xinjie Mao performed the experiments, authored or reviewed drafts of the article, and approved the final draft.
- Jiajin Wang performed the experiments, prepared figures and/or tables, and approved the final draft.
- Shouping Cai analyzed the data, authored or reviewed drafts of the article, and approved the final draft.
- Hui Chen conceived and designed the experiments, authored or reviewed drafts of the article, and approved the final draft.

### Data Availability
Raw data are available in the Supplemental Files.

## Supplemental Information

Supplemental information for this article can be found online at http://dx.doi.org/10.7717/peerj.17680#supplemental-information.

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
