# Peer review of "Response of growth and physiological enzyme activities in Eriogyna pyretorum to various host plants"

_PeerJ, doi:10.7717/peerj.17680_

## Round 0.1 · original submission · Major Revisions

The authors are requested to address the comments of all the reviewers and submit a revised manuscript.

Reviewer 1 ·

Basic reporting

-English language is clear. I noticed some mistakes and suggested corrections within "Additional comments".
-The structure of the article have standard sections. Figures are relevant with sufficient resolution. Row data have been provided. However, some of the tabs in excel file need descriptions to better understand the meaning of numbers. I also noticed some discrepancies between row data and data in the manuscript (see additional comments). Authors should check the results again.

Experimental design

-Material and Methods need additional details (see Additional comments).

Validity of the findings

In Conclusions, authors should better explain how variation in enzyme activities contribute to efficient use of different host plants.

Additional comments

The manuscript ID-92964 entitled "Response of growth and physiological enzyme activities in Eriogyna pyretorum to various host plants" assess responses of a lepidopteran polyphagous pest to three host plants that differ in chemical composition. The authors assessed biological parameters such as development duration, body size and survival at different stages of development. Also, they measured activities of digestive enzymes (amylase, trypsin, lipase), detoxification enzymes (CarE, CYP450, GST) and protective enzymes (SOD, POD, CAT) in larvae. They tried to relate enzyme activities with chemical composition of leaves. This is an original work and activities of these enzymes were not studied before in this species. My numerous comments and suggestions are listed below.


ABSTRACT
-Lines 1-4: The phrase “intricate relationship” is repeating in two sentences. In the first sentence provide more details to explain what you mean. For example, that morphological attributes and chemical composition of host plants shape growth and development of phytophagous insects via influences on their behavior and physiological processes. In the second sentence put “….studuying how feeding on different host tree species affects…..”.
-Lines 9-10: Please, avoid expressions such as “fascinating”. There are many papers showing that enzyme activities are affected by host plants.
-Lines 10-11: Why did you mention only digestive enzymes? Other examined enzymes were also affected.
-Lines 16-18: Delete “intricate”. Your results do not show clear relationship between enzyme activities and insect growth and development. Please, be more specific.

INTRODUCTION
-Line 29: “making it highly destructive” is not necessary.
-Line 40: Put space before “Therefore”.
-Line 44: Add “and other” before “pests”.
-Lines 52-53: Reformulate this sentence. Successful utilization of host plants by phytophagous insects depends on efficent nutrient digestion and defence from toxic and antinutritive secondary metabolites. Then “Firstly” and “Secondly” in the following sentences can be deleted.
-Line 55: Add “, growth and reproduction” after “survival”.
-Lines 62-63: Cited papers should be in italic and with semicolon between different papaers. Check this in the whole text. You should cite some review papers on physiological mechanisms of insect adaptations to host plants.
-Line 71: Abbreviation is AChE
-Line 73-74: Provide some examples showing that protein content affects activity of digestive enzymes.
-Lines 76-77: Delete “scientifically prevent and”.
-Line 79: Replace “varietes” with “species”.
-Line 81: Put in parentheses which parameters were recorded. Delete “meticulously”.
-Lines 82-83: Fifth instar? In line 123 you wrote that it was fourth instar. Put in parentheses which enzymes were assessed.
-Line 86: Delete “three types of”.

MATERIAL AND METHODS
-Line 91: Put “and reared” instead of “rearing”,
-Lines 93-94: Where did E. pyretorum lay eggs and how were they collected? Describe rearing box (dimensions, is it glass or plastic, etc.).
-Lines 100-101: How did you maintained freshness of leaves? Put “box” instead of “container”. Is it the same rearing box as in section 2.1.?
-Lines 101-102: It is “2nd”. Where were they kept individually? In reraring boxes or…?
-Line 103: Why were pupae collected and kept in separate containers? Pupae developed from larvae that were already separated. Describe plastic containers (dimensions? how were they covered?).
-Line 108: It is “biological”.
-Line 116: What is the duration of pupal stage? Is 14 days close to the end of pupal pupal stage?
-Lines 119-121: Add “number of” everywhere.
-Line 122: The sequence of methods should be the same as in Results. Since you first described digestive enzymes in Results you should also describe determination of digestive enzymes activity before detoxification and protective enzymes.
-Lines 123: This is not the sentence.
-Line 123: Is it 4th or 5th instar? On which day of the 4th (or 5th) instar were larvae sacrificed? Mention somwhere in section 2.4. that larvae for determination of enzyme activities were reared in the same way as larvae for determination of biological parameters.
-Lines 123-128: These sentences are written as lab protocol and should be rewritten.
-Line 131: Midgut tissues? As I understand you homogenized whole larvae.
-Line 136: Convert rpm to g.
-You did not determine protein content in homogenates and calculate specific enzyme activity. Why?
-You did not describe methods for determination of leaf composition.
-Statistical analyses: (Lines 141-142): It should be larval mass and larval length. I did not see pupal weight (mass) and pupal period (duration of pupal period?) in the Results. In Figure 2, only pupal length and pupal survival are presented. (Lines 144-145): I also did not see “the transcript levels genes associated with detoxification enzymes”. (Lines 145-146): This is not correct sentence. Mention among which data you performed correlation analyses. What was the hypothesis tested by correlation analysis? How did you test assumptions of parametric ANOVA?

RESULTS
-Line 153: Refer to Fig. 1B. I suggest that this should be the first subfugure because you first talk aboult development and then about different measures of body size. Provide data on total larval development duration and compare effects of different host plants by ANOVA.
-Line 159: In 6th instar, larval length did not differ between C.o. and L.f.
-Line 161: Add “also” after “was”.
-Line 165: Replace “smaller” with “lower”.
-Line 174: Add “and L. formosana” after “C. officinarum”.
-Line 176: Mention drastically reduced survival in 8th instar. You should also provide data on total survival during larval development and compare it between experimental groups.
-Lines 180-183: What is prepupal and larval survival? Is it survival during pupal period and total larval survival? I am not sure. Please, explain better. Total duration and survival of larval stage should be presented in Fig. 1 or in separate figure.
-Line 185: Add “larvae” at the end.
-Line 194, 201-202: Delete values of enzyme activities.
-Lines 215-218: In Table 1 it is 2.58. However, according to row data it is 1.90. Therefore, correct values in the Table 1. Also, “1.74 times” should be replaced with “1.28 times” and “3.35 times” sdhould be replaced with “2.48 times”.
-Lines 218-219: According to Table 1 flavonoid contents in C.o., L.f., P.s. are 50, 59, 53 and L.f. value is significantly higher than values on C.o. and P.s. However, according to row data values were 56, 55, 53 and may be these values are not significantly different.
-Correlation analyses: Did you determine chemical composition of leaves in leaf remains after feeding of the larvae for which enzyme activities were determined. In other words, does each value of enzyme activity in the homogenate of three larvae/midguts correspond to chemical composition of leaves that they ate? Did you collect these leaves during the larval development up to 4th (or 5th instar) or only in the 4th (or 5th) instar.

DISCUSSION
-Line 244: It should be suitability of host plants for phytophagous insects.
-Lines 256-259: Delete “intricately”. Adaptability is linked to characteristics of host plants but also on the ability of an insect to overcome various defense mechanisms of plants through behavioral and physiological adjustments.
-Line 270: Delete “higher or”.
-Lines 271-272: Explain that trypsin activity is the highest on P.s., host plant with the lowest protein content.
-Lines 274-276: Do you mean”to obtain more proteins”?
-Linesd 276-278: Unfortunately, you do not have data on lipid content in leaves and you cannot make such conclusion.
-Line 279: Delete “intricate”.
-Line 280: What do you mean with “facilitate diversity”?
-Line 281: Add “anion” after “superoxide”.
-Line 282: Which toxic substances?
-Line 285-287: Do you mean “large quantity”? Do you mean “superoxide anion radicals” or “reactive oxygen species”. You can replace “harm” with “oxydative damages to macromolecules”.
-Lines 288-289: According to row data it is not true for flavonoids. Provide some literature data how phenolics induce oxydative stress in insects and change enzyme activities.
-Line 293: Delete “harmful effects”. Laval mass, is the highest on L.f. host despite high total phenolics. In addition to POD and CAT, lipase is also the highest on L.f. Can you relate this?
-Lines 293-303: Relate data in cited papers with your data. You obtained increased CarE and CYP450 on C.o. host.
-Lines 309-311: You suddenly mentioned “intestinal microbial diversity”. So, this is a suggestion that microbiome might have important role in detoxification and should be studied.

CONCLUSIONS
-It can be seen that enzyme activities vary depending of host plants. Please, explain better which of the plastic responses are adaptive and enable higher growth and survival. Suggest further directions of research.

REFERENCES
-Line 343: Font is different.
-Line 351: “J” in journal name is not in italic.
-Lines 353, 389-390: Some authors’ names are underlined.
-Line 386: Rhagoletis should be in italic.

TABLE1: Mention nutrients before secondary metabolites. It is three not four host plants.

Reviewer 2 ·

Basic reporting

The manuscript deals with “Response of growth and physiological enzyme activities in
Eriogyna pyretorum to various host plants” deals with the research provides valuable insights into the relationship between Eriogyna pyretorum and its host plants, shedding light on factors influencing growth and development.

Comments
• The choice of three distinct host plants allows for a comprehensive examination of the insect's response to different vegetation types. The observed differences in larval development, pupal length, and survival rates highlight the significance of host plant selection in shaping the life cycle of E. pyretorum.
• How were the host plants selected for this study, and were there any specific criteria for their inclusion? What methods were utilized to measure the growth and development of Eriogyna pyretorum larvae?
• The research would benefit from a larger sample size and increased replication to enhance the robustness and generalizability of the findings. Without adequate replication, it's challenging to determine the consistency and reliability of the observed effects across different individuals and populations of Eriogyna pyretorum.
• It's important to ensure that environmental conditions, such as temperature, humidity, and light, are adequately controlled throughout the experiment to minimize potential confounding variables that could influence the results. Without proper control, it's difficult to attribute the observed effects solely to the differences in host plant species.
• How were the survival rates of larvae determined, and were there any factors besides host plant type that may have affected these rates?
• The positive correlation between total phenolics content in host plant leaves and catalase/lipase activities suggests a potential role of phenolics in influencing insect physiology.
• Could there be other factors besides host plant selection that contributed to the observed variations in developmental outcomes?
• The statistical analysis methods used should be clearly described and justified. Additionally, the significance levels and effect sizes of the observed differences between larval responses to different host plants should be reported to assess the practical significance of the findings.
• What mechanisms might explain the observed correlations between host plant phenolics content and enzyme activities in Eriogyna pyretorum larvae?
• While the study provides valuable insights into the effects of different host plants on the growth, development, and physiological enzyme activities of E. pyretorum larvae, it would be beneficial to further explore the underlying mechanisms driving these responses. Investigating molecular pathways or biochemical mechanisms could provide a deeper understanding of how host plant selection influences insect physiology.
• Can you provide insights into the potential evolutionary implications of the observed variations in enzyme activities? Were there any measures taken to control for potential confounding variables in the experimental design?
• While the study highlights the potential applications of the findings in pest management and ecological conservation, further research is needed to evaluate the practical efficacy and sustainability of implementing strategies based on these findings in real-world contexts.

Experimental design

See section 1

Validity of the findings

See section 1

Additional comments

See section 1

---

## Round 0.2 · Major Revisions

Authors are requested to revise the manuscript carefully and address all the comments.

Reviewer 1 ·

Basic reporting

I noticed some mistakes regarding English language and suggested corrections within "Additional comments". For example, some sentences are not complete.

Experimental design

Statistical analyses need corrections. Two different post hoc tests are mentioned which is not in accordance with Result section.

Validity of the findings

Conclusions still need corrections. Added sentences are not clear.

Additional comments

The manuscript text is modtly properly corrected according to my suggestions. However, some comments are not well understood and I will try to be more clear. I also have some additional comments and suggestions (see below).

ABSTRACT
-There are some repetitions in the first sentence. Change sentence into: „Morphological attributes and chemical composition of host plants shape growth and development of phytophagous insects via influences on their behavior and physiological processes.”
-Line 15: After „plants“ replace comma with „through“.
-Line 16: Replace „affects on“ with „affect“.
-Lines 16-17: Delete „By studying larvae across multiple developmental stages,” because you also studied pupae. Replace “their” with “E. pyretorum”.
-Lines 21-23: Delete „And“. „highest or significantly highest” is not clear. The sentence should be „The activities of a-amylase, lipase and protective enzymes were the highest in larvae fed on the most suitable host L. formosana which indicated that the increase of these enzyme activities was closely related to growth and development.“
-Lines 23-24: The sentence is not complete.
-Lines 24-25: Put „enzymes“ instead of „enzyme“.

INTRODUCTION
-Lines 90-92: It is not „Studies“ because you cited only one study. You can start the sentence with „Borzoui et al. (2018) demonstrated that …“. Alternatively, cite additional reference(s).
-Line 100: Delete „at each developmental stage”.

MATERIAL AND METHODS
-Line 114: Put passive voice. “Using a small spoon, the eggs were carefully scraped off from…”.
-Line 121: This is not a sentence. Also, do not start the sentence with number.
-Lines 121-124: I think that leaf petiole should be wrapped in piece of wet cotton to keep freshness for 24h.
-Lines 127-128: These two sentences are not complete.
-Line 141: It is „Prepupal stage“.
-Line 144: It is „number of pupae“ and „number of prepupae“.
-Line 146: It is „number of pupae“ and „total number of rearing larvae“.
-Lines 149-150: „Homogenates were prepared in normal saline….“.
-Line 154: „….with one larva per replicate.“ Three larvae per experimental group is not quite representative.
-Lines 151 and 159: Why centrifugation differ between sample preparation for digestive and other enzymes?
-Lines 172-173: How many leaves per sample were analyzed?
-Line 176: „…calculated for individuals fed on three host plants.”
-Line 177: Replace comma with „and“.
-Lines 178-181: These sentences are not complete. Why you used different post hoc tests for biological parameters and enzymes? You also determined correlations between enzyme activities and content of nutrients and phenolics in leaves. Did you determine correlations between mean values of traits for each experimental group? It should be mentioned. Can you also include mean values of biological parameters in correlation analyses?
RESULTS
-Line 214: You did not mention drastically lower survival of 8th instar larvae. Why is it about 3 times lower than in 7th instar? Other biological parameters only slightly differ. Total larvae duration for individuals reared on C. officinarum (Figure 2c) should be much longer than in other groups due to additional 8th instar which lasted 25 days.

DISCUSSION
-Lines 280-282: I think that both high mass and short development are usually recorded on suitable host plants.
-Lines 324-327: You should not cite other works in the sentence that describes your results. Explain in the new sentence how citied works are related to your results. Also, „….and no significant differences in the flavonoids among larvae feeding on the three host plants.” sounds like you determined flavonoids in larvae. Please, correct this.
-Lines 331-332: „The high activity of lipase in larvae fed on L. formosana leaves…”.
-Line 334: Delete „And“ and start the sentence with „The positive correlation…“.
-Line 337: Replace „….in the insect’s internal system” with “…in insects”.
-Lines 345-354: Delete „co“ in line 349. I do not understand the last sentence because you did not measure these enzymes. Please, explain better what you mean.
I see in your results that larvae fed on two suitable hosts C.o. and L.f. (high mass, short development) has different strategies of coping with different leaf chemistry (among parameters that you measured only total phenolics differed significantly – Table 1). On L.f. leaf diet main strategy is defense from oxidative stress (high POD and CAT activities), whereas on C.o. main strategy is to metabolize/detoxify allelochemicals (high detoxifying enzyme activities). On unsuitable P.s. POD and CAT activities were the lowest. Use 2-3 sentences and cite some review papers to describe how insects cope with phenolics (metabolism, excretion, sequestration).
-Lines 356-357: Can significant negative correlation between total phenolics and GST also indicate inhibitory effects of phenolics on GST? There are papers that detected inhibition in vitro and in vivo (e.g., Gawande ND, Khambalkar PB, 2014. Insect Glutathione S-transferases-key players in detoxification of insecticides. Trends in Biosciences 7:2053–2059; Tang, F., Zhang, X. B., Liu, Y. S., & Gao, X. W., 2011. Alteration of glutathione S-transferase properties during the development of Micromelalopha troglodyta larvae (Lepidoptera: Notodontidae). Journal of forestry research, 22, 447-451.; Tang, F., Zhang, X., Liu, Y., Gao, X., & Liu, N., 2014. In vitro inhibition of glutathione S-transferases by several insecticides and allelochemicals in two moth species. International journal of pest management, 60, 33-38).

CONCLUSIONS
-I do not understand „gene expression in tea geometrid“. Please explain what you mean.

FIGURES 2B and 2C: Use „prepupae“ on X axis.
FIGURE 4: In caption, mention detoxifying enzymes before protective enzymes. Also, you mention Fisher LSD while in Material and Methods you mentioned Duncan’s test.
FIGURE 5: Put „coefficients“. Mention detoxifying enzymes before protective enzymes.
TABLE 1: Use „Nutrients“ instead of „Nutrient substance“.

Reviewer 2 ·

Basic reporting

The authors made significant changes in the manuscript.

Experimental design

See point 1

Validity of the findings

See point 1

Additional comments

See point 1

---

## Round 0.3 · Minor Revisions

The authors are requested to kindly incorporate minor suggestions by the reviewer and submit the finally revised manuscript for publication.

Reviewer 1 ·

Basic reporting

/

Experimental design

/

Validity of the findings

/

Additional comments

ABSTRACT
-Line 27 and keywords: Replace all “enzyme” with “enzymes”.

MATERIAL AND METHODS
-Lines 162-165: In formulae, replace “individual” with “individuals”or delete “individual”.
-Line 193: Replace “enzyme” with “enzymes”. Please, check in the whole text. When you mention a group of enzymes with certain function use “enzymes” instead of “enzyme”.
-Lines 194-196: I suggest the following change: “We determined Pearson’s product moment correlation coefficients (P < 0.05) between the activities of protective, detoxification and digestive enzymes as well as between the growth status and enzyme activities in E. pyretorum.”
-Line 196: “We drew correlation heatmaps using Origin 2021.”

RESULTS
-Lines 306-308: Differences in activities for POD and CAT among different host plant feeding groups are similar, i.e. L.f. ˃ C.o. ˃ P.s. The same ranking can be seen for length and mass of 7th instar. So, I would expect that both POD and CAT are positively correlated to growth parameters. Please, check if CAT is negatively or positively correlated to growth parameters.

DISCUSSION
-Line 410: Put “tannins and flavonoids”.
-Line 412: Put “might suggest” instead of “suggests”.
-Lines 416-418: “…and its adaptation mechanisms warrant further investigation which should also include intestinal microbial diversity.”

-Caption to FIGURE 4: In your response, you said that you revised the text. However, you did not mention detoxifying enzymes before protective enzymes.
-Caption to TABLE 1: In your response, you said that you revised the text. However, you did not replace “Nutrient substance” with “Nutrients”.

---

## Round 0.4 · accepted · Accept

The authors have revised the manuscript as per the reviewer's suggestion. This manuscript can be accepted in its current state.